# GLOBETROTTER: UNSUPERVISED MULTILINGUAL TRANSLATION FROM VISUAL ALIGNMENT

## ABSTRACT

Machine translation in a multi-language scenario requires large-scale parallel corpora for every language pair. Unsupervised translation is challenging because there is no explicit connection between languages, and the existing methods have to rely on topological properties of the language representations. We introduce a framework that leverages visual similarity to align multiple languages, using images as the bridge between them. We estimate the cross-modal alignment between language and images, and use this estimate to guide the learning of cross-lingual representations. Our language representations are trained jointly in one model with a single stage. Experiments with fifty-two languages show that our method outperforms prior work on unsupervised word-level and sentence-level translation using retrieval.

## 1 INTRODUCTION

Machine translation aims to learn a mapping between sentences of different languages while also maintaining the underlying semantics. In the last few years, sequence-to-sequence models have emerged as remarkably powerful methods for this task, leading to widespread applications in robust language translation. However, sequence-to-sequence models also require large data sets of parallel corpora for learning, which is expensive to collect and often impractical for rare language pairs.

We propose to leverage the synchronization between language and vision in order to learn models for machine translation without parallel training corpora. Instead of learning a direct mapping between languages, we present a model that aligns them by first mapping through a visual representation. We show how vision creates a transitive closure across modalities, which we use to establish positive and negative pairs of sentences without supervision. Since the visual appearance of scenes and objects

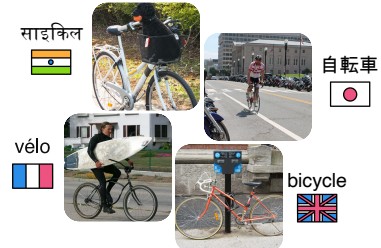

Figure 1: While each language represents a bicycle with a different word, the underlying visual representations remains consistent. A bicycle has similar appearance in the UK, France, Japan and India. We leverage this natural property to learn models of machine translation across multiple languages without paired training corpora.

will remain relatively stable between different spoken languages, vision acts as a "bridge" between them. Our approach integrates these transitive relations into multi-modal contrastive learning.

In our experiments and visualizations we show that the transitive relations through vision provide excellent self-supervision for learning neural machine translation. Although we train our approach without paired language data, our approach is able to translate between 52 different languages better than several baselines. While vision is necessary for our approach during learning, there is no dependence on vision during inference. After learning the language representation, our approach can translate both individual words and full sentences using retrieval.

The contributions of this paper are three-fold. First, we propose a method that leverages cross-modal alignment between language and vision to train a multilingual translation system without any parallel corpora. Second, we show that our method outperforms previous work by a significant margin on both sentence and word translation, where we use retrieval to test translation. Finally, to evaluate and analyze our approach, we release a federated multi-modal dataset spanning 52 different

languages. Overall, our work shows that grounding language in vision helps developing language processing tools that are robust across languages, even in cases where ground truth alignment across languages is not available. Code, data, and pre-trained models will be released.

## 2 RELATED WORK

Our unsupervised joint visual and multilingual model builds on recent progress in both the natural language processing and computer vision communities. We briefly summarize the prior work.

**Unsupervised language translation** has been studied as a word representation alignment problem in Lample et al. (2018b), where the distribution of word embeddings for two unpaired languages is aligned to minimize a statistical distance between them. Lample et al. (2018a); Artetxe et al. (2018); Lample et al. (2018c); Lample & Conneau (2019) build on top of this idea, and train an encoder-decoder structure to enforce cycle-consistency when translating from one language to another and back to the first one. This method achieves strong unsupervised word translation results, but does not scale beyond two languages. It also does not leverage visual information in learning.

**Multi-language models** are general language models that develop language-independent architectures that work equally well for any language (Gerz et al., 2018). Lample & Conneau (2019); Conneau et al. (2020); Artetxe & Schwenk (2019); Devlin et al. (2019); Liu et al. (2020); Phang et al. (2020) share the same token embeddings across different languages, showing that this improves language modeling both for general downstream single-language NLP tasks and also for supervised language translation across multiple languages. Lample & Conneau (2019); Conneau et al. (2020); Artetxe & Schwenk (2019) use a shared Byte Pair Encoding (BPE), which we use in our work. We loosely follow the architecture of Conneau et al. (2020) in that we train a transformer-based (Vaswani et al., 2017) masked language model with BPE.

**Vision as multi-modal bridge** implies using vision as an interlingua between all languages. Using a third language as a pivot to translate between pairs of languages without source-target paired corpora has been studied for the past few years (e.g. Firat et al., 2016; Johnson et al., 2017; Garcia et al., 2020). Harwath et al. (2018); Azuh et al. (2019) use vision for the same purpose, and they work directly on the speech signal instead of text. Chen et al. (2018) use images to help translate between languages in the text modality. Their model involves both generation and reinforcement learning, which makes optimization difficult, and they do not generalize to more than two languages. Sigurdsson et al. (2020) also use vision as a pivot for unsupervised translation. However, our approach works for multiple languages at once (instead of just two) and also obtains an explicit cross-lingual alignment. We share a single word embedding and language model for all languages, and use different training strategies. Our experiments quantitatively compare the two approaches, showing that our approach performs better both in word and sentence translation.

Other work views the input image as extra information for translation (e.g. Calixto & Liu, 2017; Su et al., 2019), and we refer readers to Specia et al. (2016) for an extensive overview on this topic. Instead of using images as a bridge, paired data between languages is used. There has also been research on training multilingual language representations for downstream vision tasks, in general leveraging visual-language correspondence, but without translation as a goal. Unlike this paper, they make use of ground truth language pairs (Wehrmann et al., 2019; Gella et al., 2017; Kim et al., 2020; Burns et al., 2020).

**Translation by retrieval**. We evaluate the representations using retrieval-based machine translation (Baldwin & Tanaka, 2000; Liu et al., 2012), which is often used in the context of example-based machine translation (e.g. Brown, 1996; 2001; 1997; Cranias et al., 1994; El-Shishtawy & El-Sammak, 2014), analogy-based translation (e.g. Nagao, 1984; Kimura et al., 2014), or translation memories (e.g. Chatzitheodorou, 2015; Dong et al., 2014; Wäschle & Riezler, 2015; Baldwin, 2001). While there are also generative-based translation approaches, they are difficult to automatically evaluate. There is generally no well-defined metric for what consists of a good generative translation (Callison-Burch et al., 2006). Instead, we evaluate our approach using translation-by-retrieval, allowing for rigorous experimental validation of the cross-lingual alignment in the representation.

State-of-the-art cross-lingual retrieval approaches rely on supervised language pairs, and range from training the models in a standard contrastive learning setting (Chi et al., 2020) to more complex combinations of the language pairs such as using cross-attention (Anonymous, 2021) or introducing custom fusion layers (Fang et al., 2020). Our approach does not require supervised language pairs.

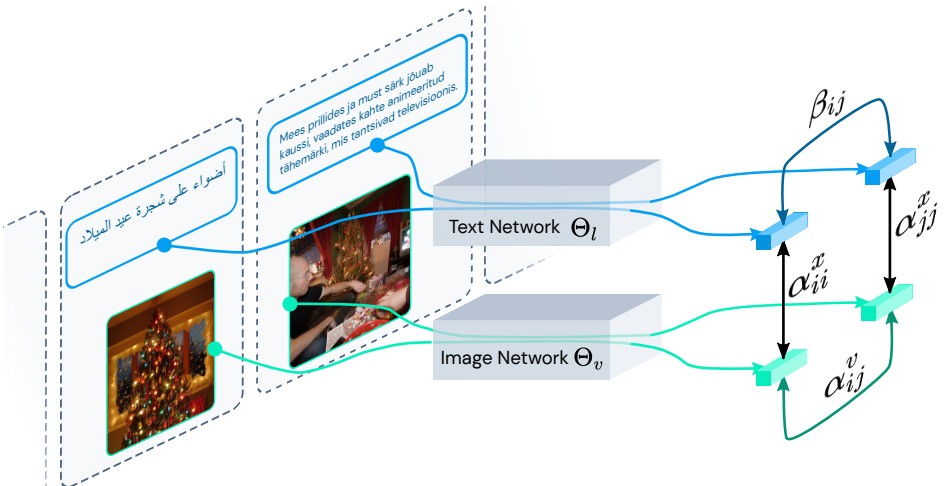

Figure 2: Our model learns an aligned embedding space for language translation by leveraging a transitive relation through vision. Cross-sentence similarity $\beta_{ij}$ is estimated by the path through an image collection. See Section 3 for details.

## 3 METHOD

We present an approach that learns to map words and sentences from one language to semantically similar words and sentences from different languages, for a large number of languages simultaneously. Our approach does not require any paired data between languages, and instead only depends on image-language pairs. Fig. 2 provides an overview of our framework.

### 3.1 SENTENCE EMBEDDING

Our approach learns an aligned embedding space for sentences across languages. Let $z_i^l \in \mathbb{R}^D$ be the learned embedding of sentence $i$, obtained by processing the text through a language network $\Theta_l$. Moreover, let $\beta_{ij}$ be the similarity between sentences $z_i^l$ and $z_j^l$, for example through the cosine similarity. Our goal is to learn the parameters of the embedding $z$ such that sentences with the same meaning are mapped to similar positions in the embedding space despite coming from different languages. After learning, we will have a sentence embedding $z_i^l$ that we can use for a variety of tasks, such as retrieving or generating sentences in different languages.

We learn the parameters of the embedding space $z$ by optimizing the contrastive learning problem:

$$\mathcal{L}_t = -\sum_i \sum_{j \neq i} \alpha_{ij} \log \frac{\exp(\beta_{ij}/\tau)}{\sum_{k \neq i} \exp(\beta_{ik}/\tau)} \quad \text{with} \quad \beta_{ij} = \text{sim}\left(z_i^l, z_j^l\right) \tag{1}$$

In contrastive learning, we need to define which pairs of examples should be close in the learned embedding space (the positives), and which pairs of examples should not (the negatives). In the above formulation, the scalar $\alpha_{ij} \in [0, 1]$ indicates this assignment. However, since we are in an unsupervised translation setting, we do not have ground truth pairs. Our main idea, which we introduce in the next section, is that we can use the visual modality to discover these pairs.

### 3.2 TRANSITIVE RELATIONS

Estimating the similarity for sentences of different languages is challenging without labels. Unsupervised machine translation approaches typically rely on topological properties, such as distributional alignment or back-translation (Lample et al., 2018b; Lample & Conneau, 2019). However, these constraints provide a noisy gradient for learning, which makes large-scale optimization difficult.

We propose to take advantage of a transitive relation through the visual modality in order to estimate the similarity in language space $\alpha_{ij}$. Given a dataset of images and their corresponding captions, we estimate both a cross-modal (sentence-image) similarity as well as a cross-image (image-image) similarity. Let $\alpha_{ii}^x$ be the cross-modal similarity, which indicates the alignment between image $i$ and its corresponding caption $i$. We also let $\alpha_{ij}^v$ be the cross-image similarity, indicating the perceptual similarity between image $i$ and another image $j$. This provides the transitive relation as the product

of similarities:

$$\alpha_{ij} = f(\alpha_{ii}^x \cdot \alpha_{ij}^v \cdot \alpha_{jj}^x), \quad \text{where} \quad f(x) = \max(0, x - m)/(1 - m), \tag{2}$$

and $m$ is a margin that we set to $m = 0.4$, which prevents pairs with low similarity from being used as positives. Note that $\alpha_{ij} = \alpha_{ji}$. The transitive similarity causes two sentences from different languages to be similar if they appear in similar visual contexts.

Since both $\alpha_{ii}^x \in [0,1]$ and $\alpha_{ij}^v \in [0,1]$, the final similarity is in the same range, $\alpha_{ij} \in [0,1]$. Only when there is a strong alignment between an image and its caption, and there is also another image with close perceptual similarity, will a transitive relation be formed. In realistic scenes, the correspondence for some image and caption pairs may be difficult to establish in the presence of noise, which our formulation handles by breaking the transitive relation. In other words, we only consider paths with high total similarity as positives for the contrastive objective, and discard those paths with low total similarity, since their sentences likely do not match.

### 3.3 LEARNING

In order to optimize Equation 1, we need to estimate $\alpha_{ii}^x$ and $\alpha_{ij}^v$. We parameterize both with a neural network, and we train them to directly estimate the similarity also with contrastive learning.

**Visual Similarity**: We jointly learn a visual feature space using contrastive learning (Chen et al., 2020) in order to estimate $\alpha_{ij}^v$. For every image, we perform two random augmentations, resulting in two different versions of the same image. These two transformed images are run through the image network, along with the other $N - 1$ pairs (in a batch of $N$ samples). This results in $2N$ feature maps. For every pair $(i, j)$ of images with representations $z_i^v$ and $z_j^v$, we compute a contrastive loss, where all the other $2(N - 1)$ images are the negatives. We use the loss function:

$$\mathcal{L}_v = -\sum_{ij} \log \frac{\exp\left(\alpha_{ij}^v/\tau\right)}{\sum_{k \neq i} \exp\left(\alpha_{ik}^v/\tau\right)} \quad \text{where} \quad \alpha_{ij}^v = \text{sim}(z_i^v, z_j^v). \tag{3}$$

$z_i^v$ represents the learned features for image $i$, obtained by processing the images through an image network $\Theta_v$. We augment images using random image cropping, random Gaussian blurring, and random color distortions, following Chen et al. (2020).

**Cross-Modal Similarity**: We also need to estimate the similarity between images and their corresponding captions $\alpha_{ii}^x$. The visual representation anchors inter-language alignment, and this similarity constrains the sentence embedding for each language to share the same space as the image embedding. We learn this similarity metric through the contrastive objective:

$$\mathcal{L}_x = -\sum_i \left( \log \frac{\exp\left(\alpha_{ii}^x/\tau\right)}{\sum_j \exp\left(\alpha_{ij}^x/\tau\right)} + \log \frac{\exp\left(\alpha_{ii}^x/\tau\right)}{\sum_j \exp\left(\alpha_{ji}^x/\tau\right)} \right) \quad \text{with} \quad \alpha_{ij}^x = \text{sim}(z_i^v, z_j^l). \tag{4}$$

**Token Cloze**: We finally also train the model with a token cloze task in order to make the language representation contextual. We follow the same loss and objective as BERT (Devlin et al., 2019) over the sentence input. We label this loss $\mathcal{L}_c$.

**Full Objective**: The final objective we optimize is the combination of all four losses defined above:

$$\min_{\Theta} \mathcal{L}_t + \lambda_1 \mathcal{L}_v + \lambda_2 \mathcal{L}_x + \lambda_3 \mathcal{L}_c \tag{5}$$

where $\Theta$ are the neural network parameters, and $\lambda$ are scalar hyper-parameters to the balance the terms. Over the course of optimization, the model will be estimating an aligned multi-lingual representation $\beta$ jointly with the transitive similarity $\alpha$. As learning progresses, $\alpha_{ij}$ will form soft positive and negative pairs, which the model will use to learn the aligned multi-language representation. The quality of the multi-language representation will depend on the quality of transitive alignments $\alpha_{ij}$ our model discovers. However, since the contrastive objective relies on statistical patterns over a large dataset, our approach is fairly robust to noise, which our experiments support.

### 3.4 REFINING WORD-LEVEL ALIGNMENT

Our approach learns a common embedding space between vision and sentences in multiple languages, which our experiments will show provides a robust representation for unsupervised ma-

chine translation. This representation is aligned well at the sentence level. We can further refine the representation by aligning them along words as well.

To obtain word-level alignment, we use the Procrustes algorithm (Schönemann, 1966) on the learned word embeddings. We find a linear transformation from the word embeddings of one language to the word embeddings of another language. To estimate the linear transformation, we follow standard practice and identify the anchor points by finding the $k = 5$ mutual nearest neighbors between the word embeddings across languages. We then proceed with the Procrustes approach from Taitelbaum et al. (2019), which extends the original algorithm to more than two distributions. To translate words, we then directly use the transformed word embeddings.

### 3.5 ARCHITECTURE

Our method uses a two-branch architecture, which extracts text and image features that share the same semantic embedding space. We briefly describe the network architecture choices below. We refer readers to the supplemental material for complete details.

**Image network**: To extract visual features, we apply a convolutional network over the images, which we label $\Theta_v$. We use a ResNet-18, initialized with ImageNet features (He et al., 2016; Deng et al., 2009), and we add a prediction head after the last hidden layer of the ResNet.

**Text network**: We use a neural network to embed a sentence, which we label $\Theta_l$. We use a single encoder with shared word embeddings across all languages, which has been shown to scale well to the multilingual setting (Artetxe & Schwenk, 2019; Conneau et al., 2020). All languages share the same vocabulary created using Byte Pair Encoding (Sennrich et al., 2016), which improves the alignment of embedding spaces across languages that share the same alphabet (Lample et al., 2018a). We then use a transformer from Vaswani et al. (2017), shared by all the languages. To produce outputs, we add a prediction head, and normalize the outputs so that $||z||_2 = 1$.

## 4 THE GLOBETROTTER DATASET

In order to train and evaluate our approach, we have collected a federated dataset of images and captions that span 52 different languages. The full list of languages is in the footnote.[1] We combined three captioning datasets and translated them using Amazon Translate from Amazon Web Services. We use captions and images from the Flickr30k (Young et al., 2014), MSCOCO (Lin et al., 2014) and Conceptual Captions (Sharma et al., 2018) datasets. The language in the federated dataset is diverse, covering both captions from human annotators and captions harvested from the web. The dataset contains a total of 4.1M image-caption pairs, with an English sentence mean length of 10.4 words. We will publicly release this dataset.

We split our dataset into a train, validation, and testing set. We make the partition ensuring that they each contain a disjoint set of images and sentences. We use 3.15M unique text-image pairs for training, 787k for validation, and 78.7k for testing. The training and validation splits contain samples corresponding to all languages, and each image only has one language associated with it. The testing set is translated to all languages (the same samples), to have ground truth alignment.

## 5 EXPERIMENTAL EVALUATION

Our experiments analyze the language translation capabilities of our model, and quantify the impact of vision on the learning process. We call our model **Globetrotter**.

---

[1]Afrikaans, Albanian, Amharic, Arabic, Azerbaijani, Bengali, Bosnian, Bulgarian, Chinese, Croatian, Czech, Danish, Dari, Dutch, English, Estonian, Finnish, French, Georgian, German, Greek, Hausa, Hebrew, Hindi, Hungarian, Indoniesian, Italian, Japanese, Korean, Latvian, Malay, Norwegian, Persian, Pashto, Polish, Portuguese, Romanian, Russian, Serbian, Slovak, Slovenian, Somali, Spanish, Swahili, Swedish, Tagalog, Tamil, Thai, Turkish, Ukrainian, Urdu, Vietnamese.

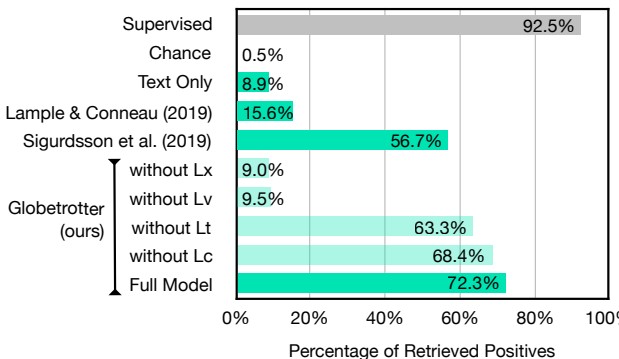

Figure 3: We evaluate our translations at the sentence-level. Our approach outperforms several unsupervised translation baselines. While unsupervised approaches are still no match for fully supervised methods, our approach uses significantly less supervision.

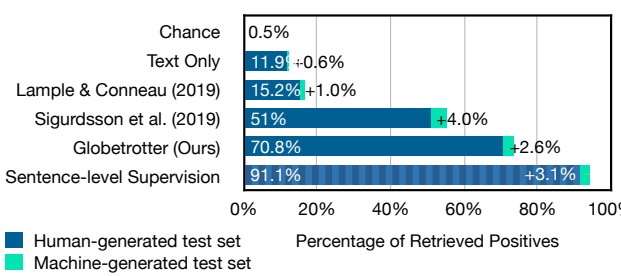

Figure 4: We evaluate our translations at the sentence-level with a human-generated test set. Fluent speakers for 11 of the languages manually annotated translations in the test set. Our approach outperforms several unsupervised translation baselines on this test set as well.

## 5.1 BASELINES

**Sigurdsson et al. (2020)**: The closest approach to ours is Sigurdsson et al. (2020), which is a state-of-the-art approach for unsupervised word translation using cross-modal information. Their original model is trained to translate between just two languages, and our experiments work with fifty languages. We therefore extended their method to multiple languages by creating a different word embedding and adapting layer for each language, which we use as the baseline. We use the same vocabulary as in our method, but train separate word embeddings for different languages.

**Lample & Conneau (2019)**: We also compare to the state-of-the-art unsupervised translation approach that does not use visual information. We experimented with several baselines, and chose the one that performs the best. This baseline uses a cycle-consistency (or back-translation) loss between pairs of languages. We train their method on our dataset, for all $M$ languages simultaneously. We originally experimented with adding cycle-consistency constraints for all $M^2$ language pairs, but this resulted in poor performance. We randomly select a total of $5M$ pairs, where each language appears five times as the source and five times as the target. We also experimented with Lample et al. (2018b), but this performed worse than Lample & Conneau (2019).

**Text-only model**: To quantify the impact of vision, we also train a version of our model where all images and image-related losses are removed, as in Devlin et al. (2019). This model is capable of learning some basic cross-lingual concepts by having different languages using the same tokens.

| Source: Spanish | Target: Russian | Target: Hebrew |
|---|---|---|
| Una vista aérea durante su remodelación 
 *An aerial view during its redevelopment* | Вид на город с бара на крыше 
 *View of the city from rooftop bar* | נוף ממרפסת גג 
 *View from a roof terrace* |
| Actor asiste al estreno de los angeles celebrado 
 *Actor attends the los angeles premiere held* | Актер посещает премьеру сезона 
 *Actor attends the season premiere* | אדם מגיע לבכורה 
 *Person arrives at the premiere* |
| Ilustración de la niña de dibujos animados en color negro sobre el fondo blanco 
 *Illustration of cartoon girl in black color on the white background* | Нарисованный эскиз с мягким классическим диваном и подушками на белом фоне 
 *Hand drawn sketch with soft classic couch and pillows on the white background* | קריקטורה של קבוצה של נערות 
 *Cartoon of a group of teenage girls* |

Table 1: We show some examples of sentence-level translations obtained by our approach. English is only shown for visualization purposes.

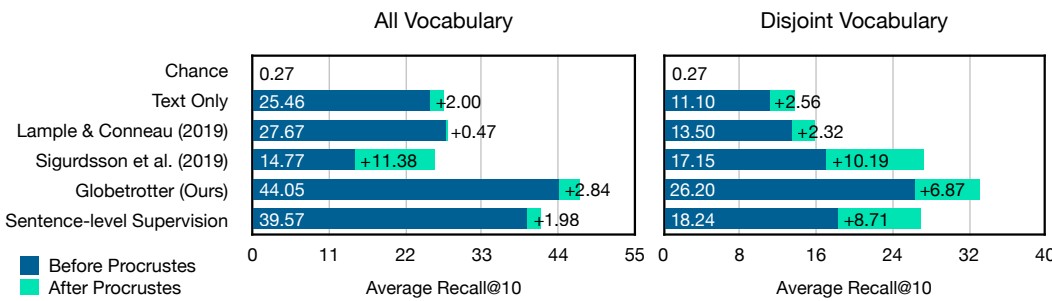

Figure 5: We also evaluate word-level translation. Although our approach is trained on sentence-level similarity, the word embeddings also learn to provide strong word-level translation. The results can be further refined with Procrustes.

| Source: Spanish (English trans.) | Target: Russian (English trans.) | Target: Hebrew (English trans.) | |
|---|---|---|---|
| chica (girl) | девушка (girl) | אישה | (wife) |
| tenis (tennis) | тенни (prefix for tennis) | טניס | (tennis) |
| personas (people) | людей (people) | אנשים | (people) |
| aire (air) | воздух (air) | רקע | (background) |
| campo (field) | поле (field) | בשדה | (in the field) |
| béisbol (baseball) | бейсбол (baseball) | בייסבול | (baseball) |
| espect (prefix for show) | шоу (show) | אירוע | (event) |
| motocic (prefix for motorcycle) | мотоцик (мотоцик is motorcycle) | אופן | (אופנוע is motorcycle) |
| camion (truck) | автобус (bus) | ברחוב | (in the street) |
| sombrero (hat) | костюм (suit) | ולצה | (חולצה is shirt) |
| hombre (man) | жчина (мужчина is man) | אדם | (man) |
| mientras (while) | когда (when) | לאחר | (after the) |
| par (two, or prefix for couple) | пара (couple) | השני | (the second) |
| calle (street) | улица (the outside) | ברחוב | (in the street) |
| camino (path) | пляже (beach) | דרך | (path) |

Table 2: We show examples of Spanish-Russian and Spanish-Hebrew word-level translations.

**Fully Supervised**: To understand the gap between unsupervised and supervised approaches, we train our method with paired language corpora. We use our same framework, except we set the values of $\alpha$ to 1 for paired sentences, and 0 for unpaired sentences.

**Common Evaluation Setup**: Throughout our experiments, we adopt a common evaluation setup to evaluate all models. We train all models for 200 epochs and select the best model on the held-out validation set. In all cases, vision is not used during testing.

## 5.2 SENTENCE-LEVEL TRANSLATION

We evaluate sentence translation using held-out data that contains a set of sentences translated to all languages. We produce translations by retrieving the nearest examples given a query. From the test set, we randomly select 200 captions, for all $M$ languages, with a total of $200M$ sentences. Each one of these sentences is used as a query during test, and it has $M - 1$ positives (same sentence in different languages). The metric we report is the percentage of positives the model ranks in the top $M - 1$, among all the $200M - 1$ possible options. In order to rank target sentences, we compute the similarity between them and the query sentence, and rank them according to this value. We show results in Fig. 3. Our method outperforms all baselines by a significant margin, underscoring the utility of transitive relations across modalities.

Fig. 3 also reports ablations of our framework when not training with each one of the four losses in Eq. 5. Training without losses $\mathcal{L}_v$ (Eq. 3) or $\mathcal{L}_x$ (Eq. 4) implies breaking the transitive closure represented in Fig. 2, which results in a drastic decrease in performance. $\mathcal{L}_t$ (Eq. 1) is the loss that makes the cross-lingual alignment explicit, but importantly it is not required to close the transitive relation through the visual modality. Training without it represents a considerable drop in accuracy, but the results are still better than baselines. Finally, $\mathcal{L}_c$ also contributes to the final performance, consistently with prior work (Lample & Conneau, 2019; Liu et al., 2020).

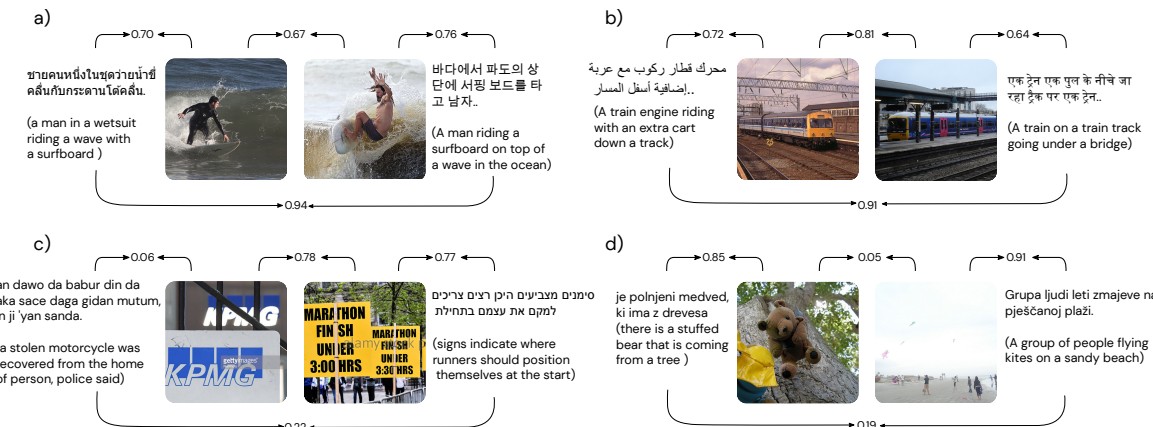

Figure 6: Qualitative results. We show two examples of positive matches (top) and two examples of negative matches (bottom). We refer the reader to Section 5.4 for further analysis.

We show some examples of our sentence translations in Tab. 1. Our approach works on all language pairs and we simply select a few for visualization purposes. These examples show how our method aligns languages following their visual semantics.

Our method does not rely on artifacts from machine-generated translations and generalizes to human-translated data. In order to prove it, we additionally collect a test set of 200 English captions translated by fluent speakers to 11 different languages, for a total of 2200 human-generated translations.[2] We report results in Fig. 4, where we show the accuracy values both for human-translated and machine-translated texts. We use the same metric as before, now for $M = 11$. Our approach outperforms the unsupervised baselines on the human-generated test as well. While all methods experience a small decrease in performance when tested in human-translated data, the small difference between the results in the two test sets validates the quality of the evaluation.

### 5.3 WORD-LEVEL TRANSLATION

Following the evaluation in Sigurdsson et al. (2020), we also evaluate the word-level translation. Since we lack ground truth translation at this level, we obtain ground truth for evaluation by automatically matching words across languages. For every language pair, we find which words co-occur frequently in a sentence between the two languages. See Appendix B.2. Then we test each pair of languages separately. For every translation, we evaluate retrieval in both directions. Fig. 5 reports the average Recall@10 for all pairs of translations and all pairs of languages. In the right column, we exclude from the list of pairs those where the token is the same in the two languages. Even the model trained with text only – which performs poorly on sentence-level translation – obtains strong results, highlighting the importance of using a shared vocabulary. We show some examples of word translation in Tab. 2.

### 5.4 ANALYSIS

**Visualizing transitive matches**: Fig. 6 shows examples of estimated transitive similarity values. We show predicted $\alpha^v$ (inter-image similarity), $\alpha^x$ (cross-modal similarity), and $\beta$ (inter-sentence similarity). Fig. 6a and 6b show examples where both the similarity between images and the cross-modal similarity are high, resulting in a large $\alpha$. If these pairs were to be used for training, they would be positives. The model correctly predicts a high $\beta$ value between the two texts. Fig. 6c demonstrates the importance of using $\alpha^x$ in addition to $\alpha^v$ to create language pairs. In this case, the visual content between the two images corresponds, and the model detects that correctly with a high $\alpha^v$ value. However, because web data is not always clean, the caption in the left does not correspond to the visual content. This is correctly captured in the small $\alpha^x$ value. If we were using this pair for

---

[2]The 11 languages with ground-truth human translations are: Dutch, French, Hebrew, Hindi, Italian, Korean, Polish, Portuguese, Russian, Spanish, Turkish.

| Source: Spanish | Target: Russian | Target: Hebrew |
|---|---|---|
| Si escuchas, el silencio de una persona te ayudará a entender de maneras que las palabras simplemente no pueden | Праздник, написанный на листе бумаги, на деревянном фоне | אם אחתוך אותך זה בגלל שנתת לי את המספריים |
| *If you listen, a person's silence will help you to understand in ways that words simply can not* | *Holiday written on piece of paper, on a wood background* | *If I cut you off it's because you gave me the scissors* |
| Un vistazo a un nuevo concepto | Заднее изображение модели автомобиля в пальто | רכישת מכונית חדשה? הנה כמה טכנולוגיות לחפש |
| *A glimpse at new concept* | *Rear image of automobile model in coat* | *Purchasing a new car? here are some technologies to look out for* |
| Un tabby gris manchado se encuentra entre plantas verdes. | Кролик ждет на переднем плане для обычной проверки | שועל אדום בשדה |
| *A spotted gray tabby sits among green plants* | *A rabbit waits in the foreground for a routine check* | *Red fox in a field* |

Table 3: We illustrate some failure cases. Please see the end of Section 5.4 for discussion.

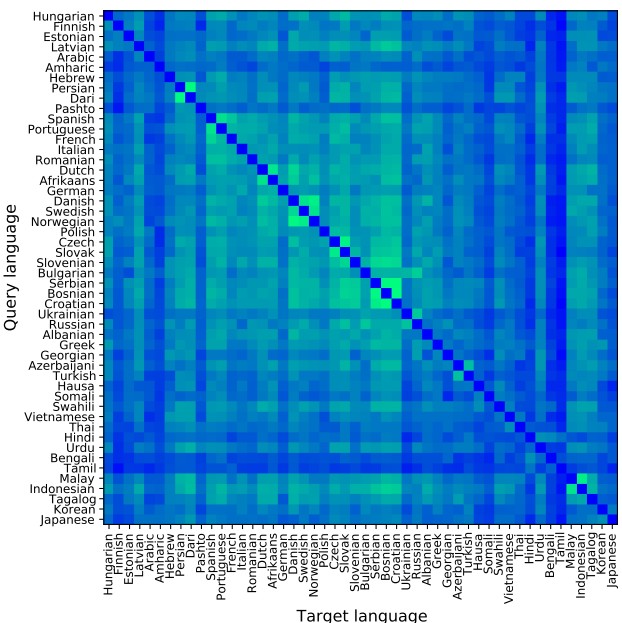

Figure 7: We show sentence-level translation accuracy by query-target language pair. In the figure, the languages are sorted by family (Romance, Baltic, etc.). The block-diagonal structure shows that languages from the same family are easier to translate between. We also find that language isolates in our dataset perform worse overall (e.g. Tamil, the only Dravidian language).

training, it would be considered a negative example despite significant visual similarity. Thus, the misalignment noise is not propagated to the cross-lingual loss. Finally, Fig. 6d shows an example where both sentences accurately describe their corresponding image, but the images do not match. As expected, this would result in a negative pair.

**Failure cases**: We show three prototypical examples of failure cases in Tab. 3. In the first example, the caption is not related to any visual concept, causing our model to translate it incorrectly. The second example shows how some words are related to incorrect concepts due to spurious correlations in the training set. In this specific case, the phrase "new concept" is strongly associated to cars, since it appears in training in the context of "concept cars", i.e. vehicles from car companies to explore new designs. Therefore, the model retrieves sentences referring to cars, even though they do not have any relation to the phrase "new concept". Finally, the third failure case shows a sentence with a new word ("tabby"), where the model is overreliant on context to translate instead.

**Translation difficulty by language**: We itemize the performance of sentence-level translation by language in Fig. 7. Languages from the same family are often easier to translate between. The most difficult language is Tamil, the only Dravidian language in our dataset.

## 6 CONCLUSION

Leveraging a transitive relation between language and vision, our experiments show our framework learns a strong representation for both sentence-level and word-level machine translation without parallel corpora. We believe vision will continue to be valuable for learning robust language models.

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

APPENDIX

We divide the appendix in two sections. In Section A we show more results, and in Section B we provide more information about the implementation of our method.

## A  ADDITIONAL RESULTS

### A.1  FEATURE GENERALIZATION

Training a language model, as opposed to a text representation only designed for image retrieval, has the crucial advantage that it can be finetuned to perform downstream NLP tasks. In this work we are interested in evaluating how well the representations generalize across languages, after training on a downstream task. We evaluate our model on sentence correspondence: we split sentences in two, and half of the times we swap the second half of the sentences with other sentences of the same language. The model has to determine whether or not a sentence is coherent and the beginning of the sentence corresponds to the end of the sentence. We control for uppercase, word breaks, length of sentences etc. so that the model cannot find an easy shortcut (cheat), and has to rely on the semantic and syntactic structure of the sentence. We show examples of the test in Tab. 4 for English.

We train all the models for one epoch on half of the languages in the testing split (first half in alphabetical order), and test on both held-out samples from that half, and *on the languages from the other half* (new languages the sentence correspondence downstream task has not seen). We train a single transformer layer on top of our representation, with one head. For Sigurdsson et al. (2020), we do not apply the max-pooling over words in order to have a representation for each word. We show results on Tab. 5. The results show that methods trained with language models are much better at performing language tasks. It also shows that our method, trained with alignment, not only performs better on the languages the downstream task has been trained on, but it also generalizes better to other languages the sentence correspondence task has never seen, indicating that the model has a very aligned representation across languages. The relative decrease in accuracy is computed as the percentage decrease of the difference between the accuracy and the chance accuracy.

### A.2  ADAPTATION TO A NEW LANGUAGE

We test how well our framework can adapt to incoming languages. For this purpose, we test on English and Chinese (separately), which were held out during training. To do so, we precompute features for images and texts from the languages we used during training, and finetune the model for the new language using the same losses as before. We train for one epoch.

After finetuning for English and Chinese, we repeat the same experiments performed for the other languages, showing that our system is able to adapt to new languages without losing the multilingual alignment. See Tab. 6 for translation results, and Tab. 7 for sentence correspondence results. For the sentence correspondence test, we use the head we trained before (without finetuning on the new languages).

### A.3  MORE RESULTS ON TRANSLATION DIFFICULTY PER LANGUAGE

Similarly to Fig. 7, we show in Fig. 8 the *word* translation accuracy matrix for every pair of languages. As expected, languages that share an important part of their vocabulary are the ones with highest similarity scores. Specifically, there is a very high similarity between Bosnian, Croatian and

| Sentence | Corresponds |
|---|---|
| *A piece of cake sitting next to pastries on a white plate with red and yellow sauce* | Yes |
| *Seamless pattern with white bugs on a black background* | Yes |
| *A big tower with a big tv genre and a common language* | No |
| *A hand holding a smartphone with of a picnic by a lake* | No |

Table 4: Sentence correspondence task examples. See Appendix A.1.

Serbian, since the three of them are standardized varieties of the Serbo-Croatian language. Also, Indonesian is very close to Malay, as the former is a standardized variety of the latter. A final example is the Czech and Slovak pair: the two of them are languages from the Czech–Slovak group. This shows the importance of cognates across languages. We can find similar patterns for languages that are not as close, but that share the same family or alphabet.

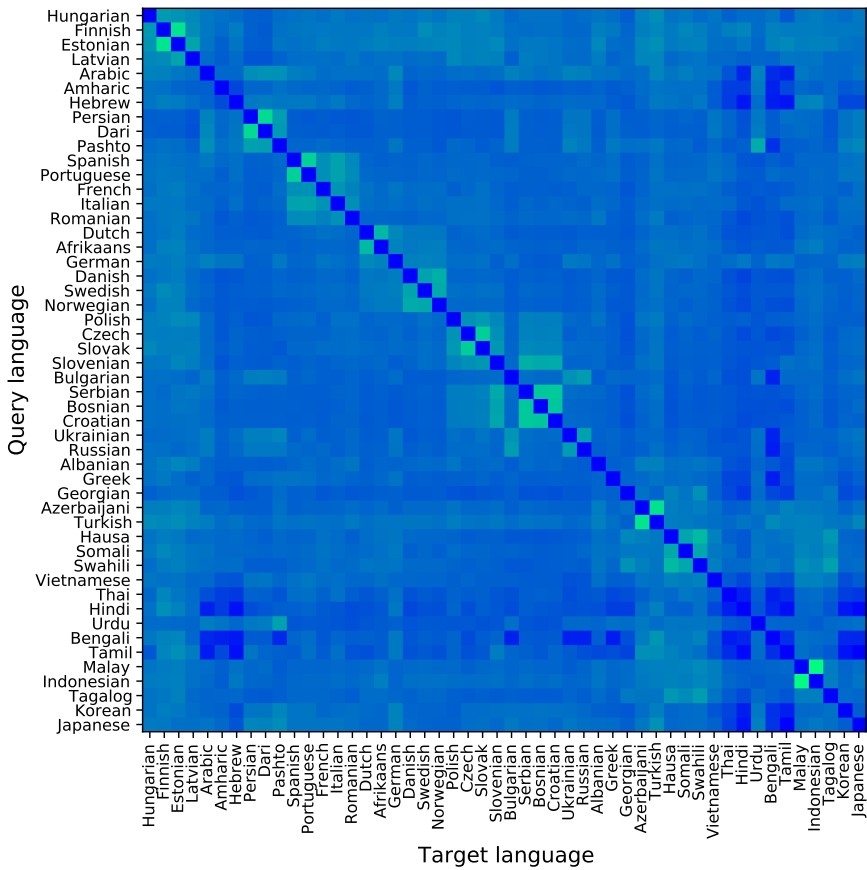

Figure 8: Word-level similarity across languages. See Appendix A.3 for more information.

We also show in Fig. 9 the sentence-level translation values from Fig. 7, but now we plot $A - A^T$. Instead of illustrating which language pairs are close, or are easier to work with, it shows which language pairs are asymmetric in the difficulty of the translation. Rarer languages —e.g. languages that are far from the others in the linguistic tree such as Somali, Tamil or Hindi— are easier to translate from than to translate to.

|  | Seen accuracy (%) | Unseen accuracy (%) | Relative decrease (%) |
|---|---|---|---|
| Chance | 50 | 50 | - |
| Text only | 71.54 | 64.94 | 30.64 |
| Lample & Conneau (2019) | 72.41 | 68.22 | 18.70 |
| Sigurdsson et al. (2020) | 53.25 | 52.89 | 11.07 |
| Globetrotter (Ours) | **75.95** | **74.54** | **5.43** |
| Supervised | 75.64 | 68.73 | 26.95 |

Table 5: Sentence correspondence results. See Appendix A.1.

|  | English retrieved positives (%) | Chinese retrieved positives (%) |
|---|---|---|
| Chance | 0.48 | 0.48 |
| Text only | 19.27 | 12.98 |
| Sigurdsson et al. (2020) | 59.18 | 37.96 |
| Globetrotter (Ours) | **75.67** | **62.81** |
| Supervised | 94.87 | 92.77 |

Table 6: Sentence translation results for finetuning. See Appendix A.2.

|  | English accuracy (%) | Chinese accuracy (%) |
|---|---|---|
| Chance | 50 | 50 |
| Text only | 65.97 | 55.75 |
| Sigurdsson et al. (2020) | 50.2 | 50.5 |
| Globetrotter (Ours) | **73.27** | **67.17** |
| Supervised | 69.17 | 62.14 |

Table 7: Sentence correspondence results for finetuning. See Appendix A.2.

### A.4 CLUSTERING IN THE REPRESENTATION SPACE

In this experiment, we show how differently the representation space is clustered when we train with and without visual alignment. We extract features for the test set examples both for the full model and the text-only model, and cluster these features using k-means, with $k = 50$ clusters. In Fig. 10 we show three sentences belonging to each one of the first three clusters (the selection of both the sentences and the clusters is arbitrary). When training with visual alignment the clusters have a semantic meaning, and when training without it the clusters are language-specific, proving that cross-modal alignment is necessary to obtain good semantic representations.

### A.5 GENERATED TRANSLATIONS

The learned representations are not only good to do translation by retrieval, but also to generate translations. In order to do so, we use a GPT-2 decoder (small version) from Radford et al. (2019), pretrained on English. Next, we finetune it on English sentences from our dataset, and after that we finetune it yet again but conditioning it on feature vectors from the English finetuned model from Appendix A.2. To do this we use an extra linear layer at the input, and we concatenate the results with the input word embeddings. After that, we obtain a GPT-2 model that generates sentences in English based on the input representation. We then test it for translation by inputting representations obtained from other languages, and generating English translations for them. The sentences we used in the test were not used for any of the GPT-2 finetuning stages. We show results in Fig. 11. We selected the first 10 translations that were generated, without any cherry-picking. Interestingly, while our framework is not able to do an accurate literal translation, it does base the translation on the contextual knowledge provided by vision.

## B IMPLEMENTATION DETAILS

### B.1 TRAINING AND ARCHITECTURE DETAILS

We train a transformer network with 4 attention heads and $M = 4$ hidden layers, with a hidden size of $d = 512$. The size of the embeddings at the output of the heads (where the contrastive losses are computed) is $D = 128$. We use a batch size of 800. We set all the $\lambda$ values in Eq 5 to $\lambda = 0.2$. We train with an Adam optimizer and a learning rate of $1e - 4$.

As mentioned in Section 3.5, we normalize the feature values $z$ so that $||z||_2 = 1$. Then the similarity value is computed with a dot product, resulting in the cosine similarity. After that, we scale the value so that the range of the similarity is in $[0, 1]$, instead of $[-1, 1]$.

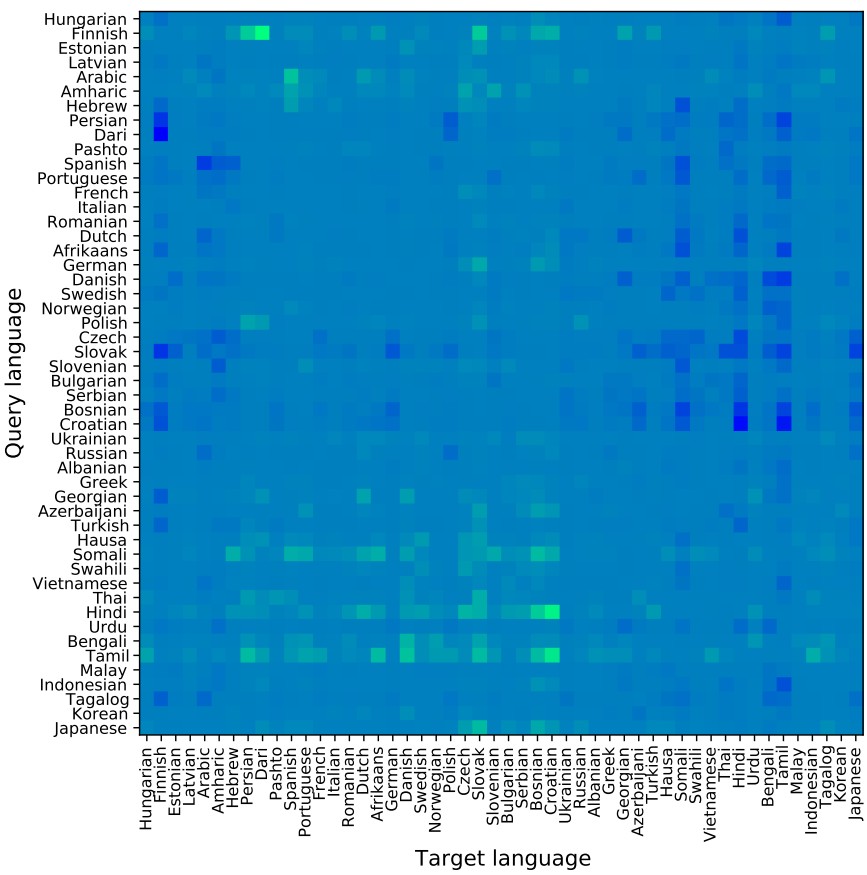

Figure 9: Asymmetry in the direction of the sentence-level translation. See Appendix A.3.

## Clusters in full model

**Cluster 1: Savannah animals**
(Arabic): يه گورخر که داره به يه گورخر ديگه نگاه ميکنه پايين يه مسير خاکي
(Croatian): popodne provedeno igrajući se sa slonovima
(Georgian): დართო გასრიოლა, ჯირაფვებს სავანის გავლით

**Cluster 2: Wedding**
(Bengali):উইন্ডোতে নববধূ এবং বর
(Slovenian): nevesta v meri obleko, ki ima roza šopek
(Urdu): !شخص کی شادی کے دن خواب سچ ہو بنانے دو

**Cluster 3: Bicycle/Motorcycle**
(Swedish): en cykel kastad ner i sanden på en strand.
(Japanese): 砂地の隣にモーターバイクが駐車しています。
(Tamil): உடற்பயிற்சி பைக் மீது பெண்.
.

## Clusters in text–only model

**Cluster 1: French**
un grand éléphant se tient près d'une clôture
motif circulaire sur fond rouge
homme silhouette à la plage

**Cluster 2: Hindi**
हाथ का एक सेट – डिजाइन के लिए प्यारा फल खींचा.
एक मॉडल घटना के दौरान फैशन शो में रनवे चलता
एक पतली परत पिज्जा तिमाही टुकड़ों में विभाजित।

**Cluster 3: Greek**
ποταμός είναι ένα δημοφιλές σημείο για κανό.
παλιά πόρτα σε ένα ξεχασμένο κήπο
πράσινα ψάρια στο γύρο ενυδρείο.

Figure 10: Clustering in the representation space. When trained without visual alignment the clusters are language-specific, and when trained with visual correspondence the clusters have a semantic meaning.

### B.2 GROUND TRUTH FOR WORD TRANSLATION

In order to generate the ground truth translations at the token level, we use the split of the dataset that is translated to all the languages. We then create ground truth token translations for every language

| Original sentence | Generated English translation |
|---|---|
| (Russian) кошка отдыхает на обочине в солнечный летний день 
(cat resting on the curb in sunny summer day) | cat lying on the grass |
| (German) Hardrock–Künstler treten während des Musikfestivals auf 
(hard rock artists perform during music festival) | artist performs on stage at festival. |
| (Croatian) Nekoliko snowboardera koji su poletjeli niz snijeg prekriveno brdo. 
(A few snowboarders taking off down a snow covered hill.) | some people skiing in the snow |
| الأوراق النقدية على خلفية سوداء (Arabic) 
(banknotes on a black background) | silver coin on the black background |
| (German) Porträt auf dem blauen Himmel Hintergrund 
(portrait on the blue sky background) | bald eagle on the green background |
| (Georgian) დამატებითი ფოტო ქონების ჩამონათვალი 
(additional photo for property listing) | photo of the front porch |
| (Swedish) utsikt över sjön från rutten 
(view over lake from the route) | picture of the beach on a sunny day |
| (Hungarian) légi kilátás a strand a legtöbb fehér és tiszta homok 
(aerial view of the beach with the most white and clean sand) | photo of the mountain lake in winter |
| (Croatian) šetnja po kiši. 
(a walk in the rain .) | photo of the rain : walking along the streets |
| (Afrikaans) akteur woon die spesiale geleentheid by 
(actor attends the special event) | some person attends los angeles premiere |

Figure 11: Translation by generation. See Appendix A.5 for more information.

pair separately. In order to do that, we follow the tf-idf algorithm. We exploit the fact that we have alignments of languages at the group-of-words (sentence) level. The idea is that if the word "car" appears in an English sentence every time that the word "voiture" (car in French) appears in its French translation, they probably mean the same. In the following explanation, assume we are looking for the translation of a specific token $t_i^A$ from language A into some token $t_j^B$ from language B. We just redefine the concept of "document" in the classical tf-idf algorithm to be the collection of all the words (with repetition) in language B that appear in the same (translated) sentence as $t_i^A$. We call this collection (document) $d$.

First, we create a count of tokens in language B that appear in the document $d$, and compute the *term frequency* (tf) using this count:

$$\text{tf}_{j,d} = \frac{f_{j,d}}{\sum_{j' \in d} f_{j',d}},\tag{6}$$

where $f_{j,d}$ is the count of the token $t_j^B$ in a document $d$. Second, we compute the inverse document frequency, that takes into account how usual a token is in general, for all $D$ documents:

$$\text{idf}_j = \log \frac{|D|}{|d \in D : t_j^B \in d|}.\tag{7}$$

Multiplying the tf and idf terms we get a value for each $(i, j)$ pairs of tokens (the value is not symmetric). We store tokens $t_i^A$ and $t_j^B$ as ground truth translation if and only if $t_j^B$ is in the top 5 for the tf-idf value of $(i, j)$, for all $j$, *and* $t_i^A$ is in the top 5 for the tf-idf value of $(j, i)$, for all $i$.

The following are some examples of translations we obtain between Spanish and English: (electr, electr), (fotograf, ograph), (ción, ction), (grande, lar), (atas, jam), (pare, couple), (decor, decor), (ventana, window), (deportivo, team), (1950, 1950), (form, form), (30, 30), (casa, hom), (lave, key), (1960, 1960), (del, the), (libro, ok), (kara, kara), (ola, surfer), (fan, fan), (viol, viol), (%, %), (dar, standard), (segundo, sec), (equipo, sports), (rojo, red), (árbol, tree), (hierba, gras), (durante, dur), (bron, ze), (mani, demonstr), (pequeño, sm), (tí, typ), (turística, attra), (corre, run), (mus, muse), (atrac, tour), (baño, bat), (mam, mom), (una, on), (element, element), (ijo, son), (ant, ol), (mural, mural), (chocola, chocola), (iste, sad), (cinta, bon), (carro, cart), (edif, bu), (planta, plant), (óc, broccoli), (prim, st), (camina, runway), (cerca, close), (pop, artist), (nacional, nation), (ustr, alian),

(vest, dress), (motocic, motorc), (perro, dog), (largo, ong), (+, +), (ates, tom), (fram, rasp), (camina, wal), (inta, inta).

## B.3 TEXT NETWORK DETAILS

The input to the text network is a sequence of tokens $\{[SEQ], w_1, \ldots, w_i\}$ that represent a sentence in any language (Devlin et al., 2019). Before inputting tokens to the transformer, we encode them with a fixed-length vector representation. To embed input tokens, we use a $\mathcal{V} \times d$ word embedding matrix $\phi_w$, where $\mathcal{V}$ is the size of the vocabulary considered by the tokenizer. We use $\mathcal{V} = 30,000$. We augment the input encoding with positional information (word index), translating the encoding by a learned vector: $\phi_{\text{txt}}(w_i) = \phi_w^T w_i + \phi_{\text{pos}}(w_i)$ where $\phi_{\text{pos}}$ encodes the word position of $w_i$.

We then input the augmented tokens to the transformer. A transformer block (Vaswani et al., 2017) consists of a multi-headed self-attention layer followed by a linear layer, that outputs a hidden representation for every token in the input sequence. These transformer blocks are concatenated in series to get deeper representations. Let $H^m \in \mathbb{R}^{d \times j}$ be the $d$ dimensional hidden vectors at layer $m$. The transformer first computes vectors for queries $Q = W_q^m H^m$, keys $K = W_k^m H^m$, and values $V = W_v^t H^m$ where each $W_* \in \mathbb{R}^{d \times d}$ is a matrix of learned parameters. Using these queries, keys, and values, the transformer computes the next layer representation by attending to all elements in the previous layer:

$$H^{m+1} = SV \quad \text{where} \quad S = \text{softmax}\left(\frac{QK^T}{\sqrt{d}}\right). \tag{8}$$

In practice, the transformer uses multi-head attention, which repeats Equation 8 once for each head, and concatenates the results. The network produces a final representation $\{h_{[SEQ]}^M, h_1^M \ldots, h_i^M\}$ for a stack of $M$ transformer blocks.

As mentioned in Section 3.5, we also add a prediction head. This head takes as input the final hidden representation for the $[SEQ]$ token, $h_{[SEQ]}^M$.

## B.4 DATASET DETAILS

To collect the dataset, we used captions from the Flickr30k (Young et al., 2014), MSCOCO (Lin et al., 2014) and Conceptual Captions (Sharma et al., 2018) datasets. Flickr30k and MSCOCO are image captioning datasets that have been carefully curated and annotated in a controlled setting, so the text descriptions are accurate and thorough. However, most of the images in our datasets come from Conceptual Captions, which consists of captions harvested from the web, so the visual-language alignment is more noisy.

We randomly split each dataset into 52 equally sized parts, one for each language supported by the machine translation service we use. Each split is assigned a unique language, and splits with the same language across datasets are combined. The split which is assigned the English language is set aside and translated into all 51 other languages, and only used in testing. We also set aside the split translated into Chinese for fine tuning experiments. The remaining 50 splits have their original English captions discarded, and are then split 80%-20% into training and validation data. All experiments shown in Section 5 are run on the reserved test data.

Note that there is no overlap at all (visual or linguistic) between the different splits, except for the test split. Please see Table 8 for more details about the dataset.

|  | Flickr30k | MSCOCO | Conceptual Captions | Total |
|---|---|---|---|---|
| Image/language pairs per language | $3.1k$ | $11.9k$ | $63.8k$ | $78.7k$ |
| Total image/language pairs | $159k$ | $616k$ | $3.3M$ | $4.1M$ |

Table 8: Dataset statistics. There are a total of 52 languages.

