# OpenReview forum: "Globetrotter: Unsupervised Multilingual Translation from Visual Alignment"
_ICLR.cc/2021/Conference — Reject_

### Official Review · AnonReviewer3 · 2020-10-14
**Interesting idea and nice empirical results but paper framing and experimental setup could be improved.**

**Rating:** 5
**Confidence:** 4

**Review:**


The authors propose a method for crosslingual sentence retrieval that uses images to ground sentences in different languages and to project these sentences into a meaningful semantic space. The data used in the model are image-caption pairs in each language, and never parallel sentences in two or more languages. Authors use existing English-language image captioning datasets (Flickr30k, MSCOCO, Google Conceptual Captions) and translate the English captions into 51 other languages using a machine translation system, therefore bootstrapping image-caption pairs in 52 languages including English. The paper is very well written and easy to read.

Some positive points: (1) the method is simple and elegant, and seems to produce strong results compared to a few other baselines; (2) the collected dataset will be released with the research community; (3) parts of the experimental setup were thoughtfully done, e.g. how to create the data splits across languages/images.

Some negative points: (1) the framing of the paper is not adequate, i.e. authors propose a model for "unsupervised multilingual translation" but they propose a model for crosslingual retrieval; (2) there are issues with the evaluation of the models, i.e. machine-translated data is used not just at training time but also as validation/test data; (3) parts of the experimental setup could be improved, i.e. a more thorough comparison to crosslingual retrieval models in the recent literature.

I recommend that the paper do not be accepted for publication in its current form for all the reasons mentioned above. I will provide detailed comments on these points below.

My first comment has to do with the framing of the paper. I am not sure I would call the proposed method one for "machine translation". This is not a generative translation model, but rather a retrieval model that retrieves similar sentences in a foreign language. I would certainly add a very big disclaimer in the introduction, if not in the title of the paper, clearly stating this. Machine translation's main issues are due to the fact a model needs to generate text in natural language, which is (1) hard to evaluate due to the difficulty in defining what consists a good translation, (2) the lack/difficulty in having adequate automatic metrics to evaluate MT, (3) the need for reference translations to which to compare hypotheses generated by the model, etc. This paper does not address any of these problems. It is true that the authors show a few examples at the end of Appendix A where they generate translations with GPT-2, but these are utterly secondary experiments, there is no evaluation conducted on the generated translations, etc.

Another central issue in the paper: Models are trained on datasets where the non-English language sides of the data were all machine-translated from English. Validation and test sets do not seem to be human translated either. To summarize: authors train and evaluate their models on data which is machine translated to begin with. Training on MT'ed data is not an issue necessarily and can often be helpful, i.e., back-translation is an example where training on additional MT'ed source sentences paired with gold-standard target sentences can help. However, you need to control for the quality of your validation and test sets. If they are also machine translated, you are probably introducing a lot of unintended biases (see [1] for a discussion). Note that in addition to evaluating in "translationese", these MT'ed data were not even validated by a human, meaning we do not even know if translations used as references in the validation/test sets are correct. This makes the whole evaluation questionable.

All issues considered, since the method is a crosslingual retrieval model that uses images, the authors should compare to other baselines proposed specifically for crosslingual sentence retrieval. One such model is X-STILTs [2], which performs well on Tatoeba and BUCC, two datasets proposed specifically for crosslingual sentence retrieval. Since these datasets have no associated images, it is not straightforward to evaluate on them with the proposed model. However, authors could retrieve images for sentences in Tatoeba/BUCC using a crossmodal retrieval model, and train their proposed model on the retrieved image-sentence pairs. Since the proposed model is already robust to noisy image-sentence alignments (Section 3.2, last paragraph), perhaps authors could show how it performs compared to strong crosslingual retrieval baselines. I mentioned [2] but there are many more, check the Google XTREME Benchmark for more examples [6].

A few other comments:

"Machine translation aims to learn a mapping between sentences of different languages while also maintaining the underlying intent." -> This is an odd way to define/introduce machine translation (MT). I'd say "underlying semantics", not intent.

"Experiments and visualizations show that the transitive relations through vision provide excellent self-supervision for learning neural machine translation" -> "In our experiments and visualizations we show (...)"

Correct citation for "Image pivoting for learning multilingual multimodal representations" is the peer-reviewed paper "https://www.aclweb.org/anthology/D17-1303/". You are also missing the paper "Sentence-Level Multilingual Multi-modal Embedding for Natural Language Processing", "https://www.aclweb.org/anthology/R17-1020/", which additionally uses learned representations for neural machine translation. Please double-check all your citations that reference pre-prints to make sure there is no peer-reviewed version of a cited pre-print available.

In your related work, perhaps you could at least mention (the massive amount of) previous work on sentence-image ranking/retrieval, even if these works are not multilingual [3,4,5]?

Section 3.2 "However, these constraints provide a sparse gradient for learning, which makes large-scale optimization difficult." What do you mean by sparse gradients?

"cross-modal similarity as well as a cross-image similarity" -> "sentence-image similarity as well as a image-image similarity"

You could improve the mathematical notation greatly. I would mention, for example, the different variables represented by alpha. The variable \alpha_{ii} is especially badly named, please use more discriminative/clearer variable names. Different indices (i, j, etc.) should index different things.

If Eq.5 is being maximised, you should not call the quantity being maximised a loss. A loss is always minimised by definition, e.g. negative log-likelihood loss.

[1] M. Zhang and A. Toral (2019). The Effect of Translationese in Machine Translation Test Sets. In: WMT 2019. URL: https://www.aclweb.org/anthology/W19-5208/
[2] J. Phang et al. (2020). English Intermediate-Task Training Improves Zero-Shot Cross-Lingual Transfer Too. In: AACL 2020.
[3] Kiros et al. (2014). Unifying visual-semantic embeddings with multimodal neural language models. In: arxiv.
[4] Frome et al. (2015). DeViSE: A Deep Visual-Semantic Embedding Model. In: NIPS.
[5] Faghri et al. (2018). VSE++: Improving Visual-Semantic Embeddings with Hard Negatives. In: BMVC 2018.
[6] Google XTREME Benchmark: https://sites.research.google/xtreme

---

> ### Author Response · Authors · 2020-11-24
> **Thank you for the valuable comments; addressing the raised concerns below.**
>
> We thank Reviewer 3 for their valuable comments, detailed reading and the highlighting of the simplicity and elegance of our framework, the strong results, the importance of the released dataset for the research community, and for describing parts of our experimental setup as “thoughtfully done”.
>
> **(1) the framing of the paper is not adequate, i.e. authors propose a model for "unsupervised multilingual translation" but they propose a model for crosslingual retrieval.**
>
> Thank you for the suggestion! Our paper builds on a large literature of retrieval based translation, so we also call our approach machine translation. For example, prior work uses retrieval for translation in the context of example-based machine translation, analogy-based translation, or translation memories, among others. Please see the last paragraph in the related work section of the updated paper and the number of references therein.
>
> In order to make this clear and up front in the paper,  we revised both the abstract and introduction to highlight that our translation is by retrieval (see last line of the abstract and last line of the third paragraph in the introduction). Following the reviewer’s suggestion, we also clarify in the last paragraph of the related work that we use translation-by-retrieval, and how that allows us to rigorously validate the cross-lingual alignment of the representations.
>
> **(2) there are issues with the evaluation of the models, i.e. machine-translated data is used not just at training time but also as validation/test data.**
>
> Thank you for raising this important point. We agree this could be a potential issue. Therefore in order to show that our method generalizes to human-translated test data, we collected a new human-translated dataset and evaluated our models on it. In short, these new results show our conclusions are still valid.
>
> We built our new test set by starting with 200 English sentences, then recruiting bilingual speakers to translate them into a target language. We translate each sentence into 11 languages: Portuguese, Russian, Italian, Polish, Hebrew, Korean, Spanish, Dutch, Hindi, French and Turkish. All of the translators are fluent in both English and the target language.
>
> We then evaluate our model and the baselines on these translations, as well as on the machine-translated translations for the exact same sentences. We report the results of our experiment on Table 4 of the updated paper, using the same metric as in Table 3, but now for $M=11$ languages. We also included an explanation of the experiment at the end of Section 5.2.
>
> Our findings still hold on this new dataset. The relative ranking between methods is maintained. (both our method and the baselines) see similar (and small) decreases in performance. Overall the results show that the human evaluation does not influence the advantage of our method with respect to the baselines, and that the evaluation gives a very good indication of how the model performs on human natural language.
>
> In the process of gathering our human-translated parallel corpus, translators remarked that the machine translations were of high quality, given the complexity of the text in the dataset. To quantify this, we computed the edit distance between the machine translation and the human translation, and the median across all languages was only 1.385 different words per sentence.
>
> The human translations will be released along with the rest of the dataset.
>
> **(3) a more thorough comparison to crosslingual retrieval models in the recent literature.**
>
> We have included a more thorough discussion on both cross-lingual retrieval methods and machine translation methods in the revised paper. However, most of these approaches learn with only one modality -- text.
>
> Instead, our paper investigates a relatively underexplored direction by leveraging images to create a transitive closure for unsupervised learning. Although there is still a gap between unsupervised and supervised methods (Figure 3), we believe our findings are scientifically important to publish. Compared to multiple established unsupervised baselines in word-level and sentence-level translations, our paper demonstrates that vision helps machine translation, substantially narrowing the performance gap. Since translation is a classical problem in natural language processing and we propose an unconventional approach, we believe this paper will receive wide interest at the conference.
>
> We already compare with XLM. Note that x-STILT, the baseline suggested by the reviewer, is equivalent to  XLM with some extra finetuning steps that depend on the downstream task. There is no specific retrieval associated with its pretraining.
>
> Finally, please note that all the “bells and whistles” of industrial machine translation systems can also be applied to our method to further refine its accuracy.

---

> > ### Author Response · Authors · 2020-11-24
> > **Addressing the minor points**
> >
> > ### Minor points
> >
> > **Improve the writing of specific sentences**
> > We modified the unclear sentences following the reviewer’s suggestions.
> >
> > **What do you mean by sparse gradients?**
> > What we meant was that they do not provide a good signal for training. We replaced it for “noisy gradients”.
> >
> > **Mathematical notation**
> > The indices $i$ and $j$ refer to the sample index, as it is standard practice in the community. $\alpha_{ij}$ refers to sentence-image similarity between the image in sample $i$, and the sentence in sample $j$. When the two samples $i$ and $j$ are the same, we use $\alpha_{ii}$. We would like to note that Reviewer 4 described the notation as “clean”.
> >
> > **You should not call the quantity being maximised a loss**
> > Thank you for pointing this out. We modified the definition of the loss so that they are now quantities to be minimized.
> >
> > **Correct citation for "Image pivoting for learning multilingual multimodal representations", and missing citation**
> > Thank you for pointing this out. We updated the incorrect references accordingly.
> >
> > We also added the citation for "Sentence-Level Multilingual Multi-modal Embedding for Natural Language Processing" in the paper.

---

> > > ### Comment · AnonReviewer3 · 2020-11-25
> > > **Small point.**
> > >
> > > Thank you for the clarifications. One small point: in the mathematical notation, if you have an $\alpha_{ij}$ variable,  $i$ indexes one thing (such as sentence position in a paragraph) and $j$ indexes something else entirely (such as word within the sentence). If $\alpha_{ij}$ and $i=j$, you simply write $\alpha_{ij}$, where $i = j$ but never $\alpha_{ii}$ or $\alpha_{jj}$. Your notation has many inconsistencies and I suggest you pay attention to it since it can really help make things more clear.

---

> > ### Comment · AnonReviewer3 · 2020-11-25
> > **Reply to rebuttal / still not fully convinced.**
> >
> > First of all, let me say I really appreciate the authors' efforts in continuing to develop their paper.
> >
> > **Regarding issue (1):**
> >
> > Although I really appreciate the authors willing to explicitly state their retrieval approach, the changes made are not enough. I don't want to sound too harsh but the entire framing of the paper, certainly the introduction, is quite misleading and still leads to confusion regarding what the proposed model actually does. You said in your rebuttal that "Our paper builds on a large literature of retrieval based translation", where "prior work uses retrieval for translation in the context of example-based machine translation, analogy-based translation, or translation memories, among others." That is really great and clear. Why don't you write that up-front in your introduction and make it clear from the start, and state how your approach builds on or differs from older work, for instance?
> >
> > In general, the changes made to the paper are adding the "*using retrieval*" string to the end of the abstract and to the end of paragraph 3, and saying later in paragraph 4 that "we use retrieval to test translation", which implies that retrieval is just done to test the model, not that the model is retrieval-based. You also mention your model performs "neural machine translation" in paragraph 3, but stating this implies you do generative translation because there are no contemporary retrieval-based "neural machine translation" models (to the best of my knowledge). When you say "unsupervised machine translation" you are also priming the reader to think of the works by Artxexe and Lample and Conneau, which are all generative models. In a  way, retrieval-based MT is an older approach (for reasons completely different from your work). There is a conflation of concepts that should be clarified from the start.
> >
> > In your related work, you say that "We evaluate the representations using retrieval-based machine translation" and "Instead, we evaluate our approach using translation-by-retrieval" but you don't *evaluate your approach using translation-by-retrieval*, your approach *is* translation by retrieval. These are very different things.
> >
> > You mention that additional results for dev/test sets that were manually collected/validated would be available on Table 4, but Table 4 has no quantitative evaluation, it only has four examples of sentences and caption "Table 4: Sentence correspondence task examples. See Appendix A.1." You probably referred to Figure 4. In the examples in Figure 4, how many confounder sentences are included together with the correct sentence to be retrieved?
> >
> > **Regarding issue (2):**
> >
> > I am okay with the evaluation you provided. I would emphasise that this is really suboptimal, though, since any machine translation paper evaluated on 200 examples would likely be ignored. However I appreciate the authors efforts and am willing to concede here.
> >
> > **Regarding issue (3):**
> >
> > I am not exactly convinced by the evaluation against crosslingual retrieval baselines. In modern NLP the task of retrieving sentences in language A given a semantically "close" sentence in language B is referred to as crosslingual retrieval in the literature. XLM is one baseline, but thinking about simple baselines you also have mBERT which is a cheap to experiment with multilingual baseline, XLM-R which is trained on much larger data and should perform much better, X-STILTS which improves on these previous models by doing intermediate-task training, etc. However, you choose to compare only to XLM, which is the older version of XLM-R. Why is that?
> >
> > I certainly agree with the authors when they say their idea is novel and has merit, and that their paper "investigates a relatively underexplored direction by leveraging images to create a transitive closure for unsupervised learning" across languages. That is indeed very interesting and exciting. But why not evaluate your proposed idea against what is already out there? Here I am referring to a vast number of crosslingual retrieval methods. They are obviously different and don't do what your method is designed to do, but they'd give a better indication of how your model compares to the literature.
> >
> > Given all the points raised, I am still not convinced the paper should be published in its current form.

---

> > > ### Author Response · Authors · 2020-11-25
> > > **Thanks for the prompt response!**
> > >
> > > Thank you for your quick reply! However, we respectfully disagree with your review. The paper addresses the problem of machine translation, so the framing is correct. The abstract and introduction clearly state it is a retrieval-based approach.
> > >
> > > Regarding the quantitative baselines, note that the paper beats the prior art on this exact problem, which is designed as retrieval based too (Sigurdsson et al.).
> > >
> > > We believe this paper makes a key contribution to the field, as outlined in your review. We believe the paper should be published for this reason.
> > >
> > > Thank you!

---

### Official Review · AnonReviewer4 · 2020-10-17
**Interesting idea and extensive experiments, with new dataset introduced that will benefit the community.**

**Rating:** 5
**Confidence:** 4

**Review:**


This paper introduces a framework that leverages transitive similarities between images and text to align multiple languages, without bi-lingual parallel training corpora. Unlike most existing unsupervised multi-modal translation works that only focus on a single (or a few) language pairs, this work proposes a one-for-many framework that can be easily applied to cover 52 languages at once.

The paper is well-written and easy to follow, and the notation is clean. The experiments are convincing with interesting case studies.

The major issue I want to raise is:

(1)The methodology proposed is retrieval-based (which is popular at the time of SMT). One shortcoming of retrieval-based methods is that its applicability is limited by the size of corpora. In terms of unsupervised translation, this problem is even more severe. Because the goal of unsupervised MT is to extend the successes of MT to low-resource language pairs. I would like to hear from the authors how we can improve or alleviate this issue.

Some other minor concerns are that:

(2)The authors evaluate word-level translation based on ground truth obtained using statistical patterns. This somehow makes the comparison with generative models (e.g. Sigurdsson et al 2020) unfair.

(3)A missing and important related work. Unsupervised Multi-modal Neural Machine Translation. Yuanhang Su et al 2020

---

> ### Author Response · Authors · 2020-11-24
> **Thank you for your positive review; addressing the raised issues.**
>
> We would like to thank Reviewer 4 for their positive review and constructive comments, and for describing our work as “well written and easy to follow”, and our experiments to be “convincing with interesting case studies”.
>
> **Low-resource languages**
>
> Thank you for raising this point. There was a typo in the paper, and we apologize for the confusion. We meant to write that our approach tackles “low resource language pairs”. We corrected the paper and modified the expression to “rare language pairs” to avoid confusion. In our method, individual languages still require enough sentences to be able to perform the translation properly. Note however that individual languages do not require any alignment with any other language, hence the use of “rare language pairs”.
>
> We believe that our data gathering method is a more scalable process, even for low-resource languages, than the traditional NLP pipeline for annotating translations. We are training on *noisy* sentences that are descriptions of images on the Internet. Many languages are low resource in NLP but have an active Internet presence allowing for gathering of this kind of data.
>
> **Minor concerns:**
>
> **Evaluation may be somehow unfair**
> The experiments are fair because they are all trained and evaluated on the same dataset and all of them use the same tokenization. If simple statistical patterns were important, all of them would benefit equally. Note that Sigurdsson et. al. is also retrieval-based.
>
> **Missing an important related work**
> Thank you for pointing this out. We updated the paper with this citation.

---

### Official Review · AnonReviewer1 · 2020-10-28
**Interesting topic but need some improvements**

**Rating:** 5
**Confidence:** 5

**Review:**

This paper introduces a framework that leverages visual similarity to align multiple languages, using images as the bridge between them. The cross-modal alignment between language and images is used to estimate and guide the learning of cross-lingual representations.

The main contribution is to apply the contrastive loss on the cross-modality pre-training. A connection or analysis to InfoNCE should be addressed. The $L_v$ is to distinguish the image and its distortion with other images. The $L_x$ is to distinguish the image and its caption with other pairs. The $L_t$ seems to be a weighted caption level InfoNCE, where weights are calculated from the visual similarity and caption similarity. The $L_c$ is actually the same as masking token loss in BERT, which can also be formulated as mutual information maximization (https://arxiv.org/pdf/1910.08350.pdf).

Since the overall loss function is a linear combination of 4 different losses, I think at least an ablation study is needed to address the importance of each loss function. Especially, I would like to see if $L_v$ or $L_x$ are removed, whether the model performance will drop significantly.

Another thing I may misunderstand is that the experiments in this paper are claimed as translation, but I think it (sentence level) is more like retrieval. I am also wondering the details of how to evaluate the retrieval task on other generative translation baselines.

Some missing references:
[1] https://arxiv.org/pdf/2002.02955.pdf
[2] https://arxiv.org/pdf/1811.11365.pdf

In summary, this paper presents an interesting topic, but the proposed method is of less novelty and the experimental design needs more improvement.

---

> ### Author Response · Authors · 2020-11-24
> **Addressing the raised concerns below.**
>
> We would like to thank Reviewer 1 for their comments. We address the raised concerns in the next paragraphs.
>
> **Main contribution**
>
> We believe there may have been a misunderstanding surrounding the paper’s main contribution. While our framework uses contrastive learning for estimating similarities, this misses the bigger picture.
>
> Our main contribution is a framework that leverages a transitive closure through vision in order to learn to do translation (see Figure 2). This transitive closure requires computing both inter- and intra-modality similarities. There are many different ways to compute the required similarity. We use contrastive loss because it has obtained very good results in the recent literature. Our framework will support any form of contrastive learning, such as the triplet loss or the ones mentioned in the review. Our contribution is broader than a new type of contrastive loss.
>
> **Ablation studies**
>
> Thank you for the suggestion. We have added ablation studies to the paper as well as additional discussion in the paper’s text. We briefly review these changes here.
>
> The results of the ablations are the following (also shown in Figure 3). The values represent the percentage of retrieved positives in the the sentence translation task:
> Full model: 72.3 %
> Without Lc: 68.4 %
> Without Lt: 63.3 %
> Without Lv: 9.5 %
> Without Lx: 9.0 %
> Without Lx and Lv (text-only baseline): 8.9 %
>
> Not training with Lx or Lv implies breaking the transitive closure represented in Fig. 2, which results in a drastic decrease in performance. This transitive closure is the basis of our approach.
>
> Lt is the loss that makes the cross-lingual alignment explicit, but importantly it is not required to close the transitive relation through the visual modality. Training without it represents a considerable drop in accuracy, but the results are still better than baselines.
>
> Finally, Lc also contributes to the final performance, consistently with prior work (Lample & Conneau, 2019; Liu et al., 2020)
>
> **Translation by retrieval**
>
> Thank you for raising this point. Our paper builds on a large literature of retrieval based translation, so we also call our approach machine translation. For example, prior work uses retrieval for translation in the context of example-based machine translation, analogy-based translation, or translation memories, among others. See the last paragraph in the related work section of the updated paper and the number of references therein.
>
> However, we agree the nomenclature is confusing! Therefore, we updated the paper to make  it clear both in the abstract and introduction that the translation we use is by retrieval. Please see the last line of the abstract and the third paragraph in the introduction. We also clarify in the last paragraph of the related work that we use translation-by-retrieval, and how that allows us to rigorously validate the cross-lingual alignment of the representations.
>
> **Evaluation of the retrieval task on the generative baselines**
>
> All the baselines, including the generative ones, encode the input sentences into a representation that lies in the cross-lingual shared representation space. We evaluate the sentence translation experiments on this representation. This representation can be fed to a decoder that generates the caption, as we do in Appendix A.5.
>
> **Missing references**
>
> Thank you for pointing this out. We updated the paper with the missing references.

---

### Official Review · AnonReviewer2 · 2020-10-28
**Great Idea for Unsupervised Machine Translation**

**Rating:** 7
**Confidence:** 3

**Review:**

The authors propose to leverage images to train an unsupervised machine translation (MT) model. Their main idea is that the similarity of images can be used as a proxy for the similarity of sentences describing the images. The sentences, in turn, can be in different languages, and knowledge about their similarity can be exploited as training signal for an unsupervised MT model, i.e., training without parallel sentences. Their model consists of a sentence encoder and an image encoder. For training and evaluation of the model, they use translations (multi-way for the test set) of image captioning datasets.


The authors evaluate on two different tasks: word-level translation and sentence-level translation. (However, the sentence-level translation is retrieval-based, i.e., no sentences are being generated.) They compare to multiple baselines, which seem to be chosen well. (The one that I would be consider missing is mBart (see below for the reference), but that's hard to train on many common GPUs, so I wouldn't necessarily expect that.)

Overall, the idea the authors are presenting is convincing, the experiments are clear and well designed, and the model makes a lot of sense. I don't see any major shortcomings of this paper. Thus, I would be excited to see it being presented at ICLR.

A minor shortcoming is that mBart isn't mentioned anywhere. In fact, the authors claim that they propose the first multilingual unsupervised model (lines 3 and 4 in the abstract). This should be corrected. The reference for mBart is: Yinhan Liu, Jiatao Gu, Naman Goyal, Xian Li, Sergey Edunov, Marjan Ghazvininejad, Mike Lewis, and Luke Zettlemoyer. 2020. Multilingual denoising pre-training for neural machine translation.

---

> ### Author Response · Authors · 2020-11-24
> **Thank you for your positive review and suggestions.**
>
> We would like to thank Reviewer 2 for their positive review and constructive comments! We are excited that you found the paper to be “convincing”, with “clear and well designed” experiments, and with a model that “makes a lot of sense”.
>
> **mBART is not mentioned.**
>
> Thank you for noticing this missing reference. We apologize for missing it, and we added it to the updated version of the paper.
>
> Note that mBART is conceptually similar to XLM (Lample & Conneau, 2019), which we already report and compare against. Both methods do not exploit the image alignment that proves crucial in our setting. Also, the improvements in mBART with respect to XLM can be also applied to our method, making the two methods complementary.
>
> Thank you for your suggestions on the abstract. We have integrated them into the updated version of the paper.

---

### Author Response · Authors · 2020-11-24
**Submission Update**

We would like to thank the reviewers for their constructive feedback and suggestions. We are glad all the reviewers generally found our paper to be convincing, simple, and elegant, with strong empirical results and interesting case studies. Although machine translation is a classical problem in natural language processing, our paper introduces an alternative approach. We show how images establish a transitive closure across modalities, which we leverage to learn a cross-lingual representation without parallel training data. Experiments on 50 languages demonstrate the advantage of our approach for both word and sentence level translation. Due to the popularity of this problem and our unconventional approach, we believe this result will receive substantial interest at the conference.

The reviewers made several suggestions, which we have integrated into the revised paper. In addition to individual responses to reviewers, we briefly summarize the new additions below:
- Collection of a new dataset with human-translated test data. We show results in Figure 4.
- More ablation results, for each of the losses. We show results in Figure 3.
- Additional paragraph on related work, providing context for the use of machine translation by retrieval, and more thorough discussion on machine translation methods and cross-lingual retrieval methods.
- Adding missing references pointed out by reviewers.
- Other minor corrections.

---

### Decision · Program_Chairs · 2021-01-07
**Final Decision**

**Decision:**

Reject

**Comment:**

Overall, all reviewers generally agree that the idea of using visual similarity to unsupervised alignment of multiple languages is interesting and the proposed method and dataset are well-designed, while three of them raised some concerns related to the retrieval nature of the method. In particular,  discussions about its place as a study of machine translation and comparison with other cross-lingual retrieval baselines were the main issues. Although authors made great effort to address reviewers' concerns points and did clarify some of them, unfortunately the reviewers were not fully convinced by the response, and one reviewer decided to downgrade the initial score.  After all, three reviewers rate the paper as 'below the acceptance threshold'. Based on their opinions, I decided to recommend rejection.

I think the entire picture of the work and the logic flow could be much clearer by discussing in a top down manner why this idea should be implemented with a retrieval-based approach, rather than superficially adding "using retrieval" to some sentences.